# Manipulations of Oral Medications in Paediatric Neurology and Oncology Care at a Swedish University Hospital: Health Professionals’ Attitudes and Sources of Information

**DOI:** 10.3390/pharmaceutics13101676

**Published:** 2021-10-13

**Authors:** Rania Kader, Gunnar Liminga, Gustaf Ljungman, Mattias Paulsson

**Affiliations:** 1Department of Women’s and Children’s Health, Uppsala University, 751 85 Uppsala, Sweden; rania.kader92@gmail.com (R.K.); gunnar.liminga@kbh.uu.se (G.L.); gustaf.ljungman@kbh.uu.se (G.L.); 2Division of Pharmacokinetics and Drug Therapy, Department of Pharmaceutical Biosciences, Uppsala University, 751 85 Uppsala, Sweden

**Keywords:** paediatrics, children, manipulation, oral medication, patient safety, survey, health care professionals’ attitude, enteral feeding tubes, pharmacoprinting

## Abstract

Oral administration of medications to children requires age-appropriate dosage forms and strengths. In this study, we: (i) assessed the extent of oral dosage form manipulations, (ii) documented how it is carried out, and (iii) examined the attitudes and sources of information regarding the handling from healthcare professionals. Prospective reviews of electronic records, ward observations, and clinician surveys were performed at a paediatric neurology ward and a paediatric oncology ward in Sweden during April to May of 2018. Approximately 15% of oral medications were manipulated for the studied patient group (median age 12.9 years in oncology, 5.8 years in neurology) with approximately 30% of the patients having an enteral feeding tube. Manipulations were performed both to obtain an appropriate dose from, for example, a fraction of the original tablet or to obtain a powder that could be used to prepare a slurry for administration through enteral feeding tubes. Risks identified were related to patient safety such as cross contamination, suboptimal absorption/pharmacokinetics and inaccurate dose. When examining the working environment of nurses, we observed safe handling of hazardous substances but the nurses occasionally experienced stress and a fear of making mistakes due to absence of information. Paediatricians experienced a lack of time to search for proper information on manipulations. As a step towards improving safety in paediatric medication, we suggest the introduction of clinical pharmacists into the team and further evaluating the possibilities of using more ready-to-administer medications with necessary product information and pharmacovigilance support.

## 1. Introduction

Medication dosing in paediatric patients is challenging due to the extensive variation in patient weight in addition to changes in metabolic capacity, distribution sites and organ function [1]. For intravenous medications, scaling of dose based on, for example, body surface area or body weight is often feasible, but for oral administration, paediatricians frequently resort to licensed, fixed-dose tablets, developed for the adult population. Some oral solutions are marketed that allow for on-label dose scaling and improvements in availability are being made [2], but challenges persist [3]. The increased need for precision medicines, also including patient groups other than children, may also intensify the demand for dose adjustment outside the dose bands currently available. It can also be noted that electronic health record (EHR) software often contains a computerized physician order entry (CPOE) module that allows for easy dose scaling of orders based on a mg/kg approach that relies on the availability of scalable formulations such as oral suspensions for safe administration.

Splitting tablets has been a common practice for obtaining the prescribed dose when the specific dose is not available as a licensed product. Furthermore, crushing of tablets is a means of easy administration through nasogastric tube (NGT) or percutaneous endoscopic gastrostomy (PEG). To describe this process, the term manipulation of medications has been defined [4] and further discussed by others [5,6,7,8,9] on the basis of the guidelines from the European Medical Agency Paediatric Committee (PDCO) [10].

The aim of this study is to: (i) assess the extent of paediatric in-patient oral dosage form manipulations, (ii) document how the manipulation is carried out, and (iii) examine the attitudes and sources of information regarding the handling from healthcare professionals (HCP).

## 2. Materials and Methods

### 2.1. Prevalence of Manipulated Dosage Forms

To study the prevalence of manipulated oral dosage forms, a prospective study at Uppsala University Children’s Hospital, Sweden, was conducted. The hospital is a tertiary referral hospital with many paediatric specialists including being one of Sweden’s six regional paediatric oncology centres. The two paediatric wards—ward A (paediatric blood and tumour diseases, 12 beds) and ward B (paediatric neurology, 8 beds)—were selected, as they provided a context of specialized care with extensive medicinal treatments. The paediatric patients were included based on the following three criteria:(1)Admission to ward A or to ward B.(2)Patients aged 18 months–18 years. This inclusion criterium was set to also include young children, based on the finding that children, even as young as one year, may accept a solid oral dosage form [11].(3)Patients who were administered oral medications.

The study period was 12 March 2018 to 20 April 2018, 31 days in total. Manipulations were defined as physical activities to modify the dosage form (e.g., crushing or dividing) (i) to obtain an appropriate dose, e.g., a fraction of the original tablet, or (ii) to obtain a powder that could be used to prepare a slurry and given through enteral feeding tubes or gastrostomy. Manipulations that were not described as applicable in the Summary of Product Characteristics (SmPC) were considered off-label. Missed doses and paused medication therapies were not taken into consideration during the review. Extemporaneously compounded medications from a pharmacy or imported medications (licensed in other countries) were not considered to be a manipulation in this study. All manipulations of medications were performed by nurses in the ward medication rooms (equipped with standard tablet splitting/crushing apparatus and negative pressure safety cabinet for handling of hazardous, e.g., antineoplastic medications). The methods used in this study were reviews of electronic health record (EHR/CPOE) in the software Cosmic (Cambio Healthcare Systems AB, Stockholm, Sweden) [12]. Cosmic supports both medication orders for outpatient and inpatient care and to study the inpatient care, the data collected was specified to dose administration occasions rather than orders/prescriptions. Cosmic provides time of order and planned time of administration for medication products (no generic prescription) and is the software tool used in wards both for paediatricians’ order and nurses’ medication administration. Data about manipulations of specific medications were gathered from Cosmic and compiled in Microsoft Excel using a patient code to protect the patient confidentiality. Active substances were classified using the Anatomical Therapeutic Chemical (ATC) classification system [13] and used for further analysis.

### 2.2. Ward Observational Study

One researcher, with a pharmacy degree, was present for observation of the medication handling of nurses on the wards in the equivalence of one full day per ward (office hours) in March and May 2018. The researcher followed and observed one or several nurses closely during the study day to see the practical side of how manipulations were handled. Observations were noted in free text form. The objectives of this part of the study were for the researcher to understand the information flow and the clinical context of the CPOE information as well as to capture aspects that were not covered elsewhere. Nurses were informed about the study beforehand on ward meetings and through email.

### 2.3. Survey Study

The development of the two questionnaires (one for paediatric nurses, one for paediatricians) was based on a literature study concerning manipulations and the current situation of medication in paediatric care [10,11]. A panel based on the authors, ward management and a senior paediatrician (focused on patient safety and the development of hospital medication handling) comprised the survey validation team. A pilot testing to check feasibility and logic was performed on a staff meeting where six nurses from ward B read and commented on the survey form. This meeting also included novices such as newly employed junior nurses.

The survey comprised a series of closed-ended questions on 3-point and 5-point Likert scales. The option to add information (open-ended) was also possible for some questions.

The inclusion criteria were: (1) specialist doctors in paediatric neurology or paediatric blood and tumour disease; (2) paediatric nurses on the two wards.

Survey dissemination was done by sending a link to the questionnaire on Microsoft Forms to the nurses (using their work e-mail addresses) administered by their respective head of ward. The doctors received the questionnaire on paper on staff meetings by the two paediatricians involved in the project, one from each ward. The answers were transferred into Microsoft Forms in order to obtain a compilation of the results. Microsoft Excel was used for calculations and graphical presentations.

The authors followed the Equator network recommendations and used the paper by Kelley et al. to report results [14]. The survey was written in Swedish and translated to English by the author team, including a discussion on content validity.

## 3. Results

### 3.1. Prevalence of Manipulated Dosage Forms

The administered oral medications were given to 79 patients, predominantly older than 2 years with a median age of 12.9 years (Oncology, ward A) and 5.8 years (Neurology, ward B), Table 1. The patient weight in this study ranged from 6.7 kg to 109 kg, which corresponded to a 16-fold variation. Many of the patients were referred to Uppsala for treatment, with only 34.8% of Oncology patients and 54.3% of Neurology patients living in Region Uppsala. For the Oncology ward, patients came from all over Sweden because of the neighbouring national proton therapy centre, the Skandion Clinic.

During the observation period, a total of 330 patient study days were analysed, with 2042 active, inpatient orders, and a proportion for oral delivery of 62%. For the 1358 administered oral medication doses in the two wards, the majority (70%) were given to adolescents, with only smaller numbers (20% for schoolchildren, 8% for pre-school children and 2% for toddlers) given to younger patients.

Manipulation of dosage forms due to the absence of appropriate strengths occurred in 10.8% of the active oral medication administrations for the paediatric oncology ward and 9.1% for the neurology ward (Table 1). Tablets were the main dosage form that was manipulated (94%), followed by adjusting quantity of sachet content to obtain correct dose. In three instances, modified release (MR) dosage forms were also manipulated, videlicet bisected (Table 2).

Enteral feeding tubes or gastrostomy were present in 26.4% (ward A) and 31% (ward B) of the patients. It was not possible to find information in the CPOE for each administered dose whether a tablet was given orally or via the tube. Based on discussion with nurses, it was assumed that for a patient with enteral feeding tube/gastrostomy, all conventional tablets are given using the tube. Manipulations for purposes of facilitating administration was 2.2% for ward A and 11.2% for ward B (Table 1).

Manipulations, both for obtaining a proportion of the dosage form, as well as modifying to facilitate administration, were most common for toddlers (64% of all doses) followed by pre-school children (52%), schoolchildren (13%) and adolescents (11%) resulting in the overall fraction of manipulation being 15.5%.

Patients (or parents) had accepted the responsibility to self-administer (without the presence of nurse) some or all oral medication in 11.9% or 10.1% of cases for ward A and B, respectively.

### 3.2. Ward Observational Study

Findings from the ward observation period are summarized in Table 3. It is evident that there are multiple methods in the wards for preparing a dose based on a fraction of a tablet. Manipulation of tablets induces risks both regarding patient safety and nurses’ working environment.

### 3.3. Survey Study

We analysed responses from 20 specialist paediatricians collected in April 2018, with 85% response rate. Respondents were consultant specialists in paediatric blood and tumour disease (response frequency 8/11) and paediatric neurology (response frequency 9/9). The survey of the 62 paediatric nurses had a lower responding frequency, 34%, distributed as 30% on ward A (9/30) and 37.5% on ward B (12/32).

#### 3.3.1. Paediatricians’ Perspective

The results from the survey focusing on information availability and practical handling is presented in Table 4. Regarding the question on HCPs ensuring that the medication order is compliant with the national paediatric regulations [15] (stating that the child’s weight/body surface/age and the strength of the diluted medication in addition to the maximum dose of the medication should be stated in the order), 58.8% say that they are always/often sure this is met. A proportion of 18.8% of paediatricians agree that the CPOE systems are insufficient to handle documentation of manipulations but paper-based medication orders was not a commonly used alternative. The paediatricians experienced that they rarely or never (52.9%/17.6%) had the time to search for proper information in order to write a thorough procedure for the manipulation subsequently executed by nurses (Table 4).

According to the respondents, the ‘FASS’ web page [16], containing the summary of product characteristics (SmPC), was mainly used to find information regarding whether tablets could be split/crushed or formulations manipulated, but other sources were also used (Figure 1). Paediatricians stated they did not have enough time to search for information or to write thorough instructions for the manipulations.

#### 3.3.2. Nurses’ Perspective

The results from the survey focusing on information availability, practical handling as well as work environment is presented in Table 5 and Figure 2. Patient safety and experience of adverse events were also included. The survey likewise examined the nurses’ opinion on ready-to-administer (RTA) medications. The nurses answered that clear instructions about manipulations could rarely be obtained in the CPOE, and that there were rarely or never clear instructions on the execution of the manipulation in the medication order.

The nurses stated that they had to manipulate the dosage form of medication occasionally or often, and the main complication was considered to be occlusion in the feeding tube. When complications occurred, they were documented in the patient medical records and the information was shared with colleagues at meetings.

To gain information regarding the procedure of a manipulation, the main source of knowledge was to receive instructions from a colleague. The nurses thought that they often had the time to search for information of the procedure of a manipulation. Nurses experienced it to be stressful to perform manipulations, even though they occasionally or often had the time to execute them.

The CPOE rarely (38.1%) contained clear instructions for acquiring information on a procedure of manipulation. When the CPOE did not provide sufficient information, the main source used was the SmPC and patient information leaflets (Figure 1). It was clear that nurses would appreciate to receive RTA medications from the pharmacy.

Occasionally, conflicts or discussions emerged between colleagues due to differences in opinions regarding procedures. It happened that medications were mixed with food, the nurses rarely reflected upon the potential risks in doing so, but there are interindividual differences depending on level of experience. The work by nurses is heavily reliant on how thorough the instructions are from the paediatricians. In Table 4 it can be seen that only 11.8% of paediatricians often had the time to write thorough instructions on manipulations.

## 4. Discussion

### 4.1. Prevalence of Manipulated Dosage Forms

The patient cohort showed a wide range of body weight, 16-fold, which is often seen in paediatrics and is a challenge related to medication handling errors. Many of the patients were referred to the hospital from other regions and this requires further distinctions in information transfer between hospitals, hospital CPOEs and for handling at home by parents or caregivers.

Around 1/10 of the patients exerted “self-administration”, that is, they administered some or all oral medication by themselves and the nurse documents in the CPOE only that the medication was delivered to the patient (or parents) and not the actual administration. This helps in engaging the family in the treatment of the child and improves concordance after discharge but may increase risks if extensive manipulations are needed.

The medication lists of many children were rather extensive since the CPOE also contains medications that are used at home. Many of these medications are temporarily withdrawn or exchanged for intravenous alternatives during the hospital visit, in addition to PRN (*pro re nata*) alternatives. This results in only part of the orders being active in the hospital. Around 15% of all medications in this study were manipulated, and this figure is similar to the findings of others [5]. Manipulations are more common in younger children and depending on age of the studied group or ward specialisation, numbers could vary greatly. Determined by how extemporaneous compounding is classified also higher figures of manipulations, e.g., 37% are seen [7].

Several of the substances in Table 2 are high-alert medications (i.e., has a heightened risk of causing significant harm to patients) as stated by the Institute for Safe Medication Practices [21]. These medications have since been examined on paediatric drug-related incident reports and complaints highlighting alert awareness as a potential medication error reduction strategy [22]. Some of the substances, e.g., valproic acid, are licensed in Sweden as an oral solution but this product was not used for the patient dose described in Table 2. The reason behind this is not clear; it could be due to HCPs being unaware of the product, a temporary drug shortage or most likely that a sustained release profile was desired but needed only a fraction of the dose.

Bisecting modified release tablets, as was observed for metoprolol, felodipine and valproic acid in this study, is something that obviously must be advised against. The risks associated with splitting MR tablets (i.e., metoprolol) or MR capsules (i.e., valproic acid) containing MR granules are likely not as severe as those seen when manipulating MR tablets based on an outer MR tablet coating. Nor should tablets be split more than once (baclofen in Table 2) or other partitions (hydrocortisone, phenytoin, clobazam in Table 2) due to uncertainty of dose accuracy [23]; however, both tablet formulation and dose extraction technique must be considered when manipulations are done [24,25].

RTA oral suspensions/solutions may for some medications be available from extemporaneous compounding pharmacies or by importing from other countries but due to short shelf-life/long delivery time, this is often not an achievable alternative for medications rarely used in hospitals with small patient bases. During the time of the study no pharmacy was able to provide compounding services for non-sterile chemotherapy based on tablets as raw material.

### 4.2. Ward Observational Study

The dose accuracy problems when preparing a dose based on a fraction of a tablet, described in Table 3, are particularly severe for active substances with a narrow therapeutic window. The division method based on dissolution assumes high aqueous solubility of the active substance and since this is often not the case (in addition to presence of insoluble tablet excipients) this method is also uncertain and could give incorrect dose or different dissolution/absorption profiles [24].

Around 30% of the patients in this study have feeding tubes (Table 1), and it is hence convenient to crush all solid oral medications to achieve a higher compliance, but this likely increases the risk of variations in pharmacokinetics due to differences in absorption. Crushing of tablets is also done for patients without tubes in order to allow for ease-of-swallowing, with tablets being dispersed in a slurry for the patient to drink. It should be noted in this context that no documentation exists in the CPOE as to whether the patient swallowed the tablet whole, drank a slurry or if the administration was done using NGT/PEG. This level of detail is likely not required by authorities in many countries, but would be informative for treatment follow-up. Mixing medications with food may increase the risk of food refusal, as described in Table 3, but may also affect the absorption kinetics of the active substance. The tablet crushing device used during the study period was, as described earlier, difficult to clean with the potential loss of dose in handling. The device has, since the study was conducted, been replaced by a more appropriate device in which the tablet powder only is in contact with single use disposable material during the crushing process. The findings in Table 3 could be seen as examples for others to use as quality improvement projects based on the mandate from the EU resolution on good reconstitution practices [26].

### 4.3. Survey Study

Nearly half (49%) of all administered prescriptions in Swedish paediatric hospitals concerned unlicensed drugs, off-label drugs or extemporaneously prepared drugs [27,28,29], often associated with limited handling instructions from the supplier. The paediatricians stated that they rarely had the time needed to search for information on the procedure required for manipulations. In addition, they experienced lack of time to write thorough instructions, time that increases when they are forced to search for information that is not easily available, but much needed. The working environment is of great importance in ensuring good practice of care. The HCPs overall experienced stress, disturbance in their work and lack of time. These are important factors which, if reduced, could improve medication safety of patients [27].

The nurses experienced the manipulations as stressful to perform, despite the fact that they believed they had the time to execute the procedure. Perhaps time is not the limiting factor, but rather the insecurity of whether or not the manipulation will be correct. This finding is outside of what is typically referred to as work-related stress in nursing [30] and would be interesting to study further. Nurses also said they often had the time to search for additional information on how to perform a manipulation when there was no instruction in the CPOE, which often appeared to be the case. Thanks to the safety cabinets of the medication room, exposure to hazardous substances during manipulation was minimised for nurses. Approximately half of the paediatricians perceived it to be difficult to find information to write thorough instructions (Table 4).

In this study, it was also found that the state-of-the-art medication order software used still lacks sufficient decision support for, and documentation of, the performed manipulations. We realized that it was not possible to retrospectively ascertain in which way a tablet was manipulated and if it was administered through a tube or not. Multiple procedures for preparing a dose based on a fraction of a tablet were also evident in the two wards studied. These findings would likely surprise many formulation scientists or medication safety officers from the pharmaceutical industry that are responsible for the medication product license. To date, many countries and territories have adopted EHRs and associated digital technologies for safe medication handling, but progress toward a transformation of digital health systems to be more proactive, predictive, high performing and focused on supporting population health and wellness has been limited [31]. Based on a European Union resolution [26], a paediatric best practice initiative is now in place in Sweden [20], covering recommendations and training in the handling of medications for children. This information database is continuously expanded and integrated with many Swedish EHR software systems.

### 4.4. General Discussion

Medication safety is important for hospitalised children, as they are at risk of experiencing unintended harm as a result of medication errors. There is only a handful of medication safety strategies (e.g., clinical pharmacists, CPOE, barcode scanning) that have been studied using robust study designs. In a Cochrane review of studies from earlier than 2014, no clear evidence for a specific strategy was found [32] however in a more recent review study, evidence supported the implementation of clinical pharmacists in order to reduce the occurrence of medication errors in paediatric patients [33]. Pharmacist interventions are effective for reducing medication error rates in hospitalized paediatric patients [33,34], and their knowledge is likely especially valuable when finding safe products, preventing interactions when prescribing or incompatibilities when administering as well as gathering and presenting handling information for use in wards, at home or when the patient is transferred to another, often smaller hospital with fewer resources. The authors are happy to share the information that for Ward A, a full-time clinical pharmacist was permanently employed 18 months after the study was performed, and more pharmacists are planned to join the unit.

There are also other good initiatives [10,15,20] at different levels (regulatory authorities and health care organizations) to reduce the risk of errors but it is likely that patient safety, working environment for nurses, treatment follow-up, etcetera, could be improved with more child-appropriate licensed medications or extemporaneous medications. A prerequisite for extemporaneous RTA is a shift towards centralised preparation with prompt delivery, likely within 2–12 h of time of prescription. This could be achieved through small-scale local capsule filling or through the development of on-demand pharmacoprinting based on additive manufacturing (2D/3D) [35,36,37,38], where other means (e.g., colour, taste, shape, release) of age appropriateness can also be used. Key formulation attributes can be managed using a paediatric Quality Target Product Profile [39] and by including unique identifiers, e.g., 2D barcodes, at the individual dosage unit giving an easy access to tailored information regarding the drug product [40]. There is still likely a great need for further development of the pharmacoprinting technique, including formulation science, before it can be realised in the hospital pharmacy.

The strengths of this study are the important subject of paediatric medication safety and that we have examined the attitudes of HCPs in a tertiary hospital. Weaknesses are the limited number of patients and the limited number of HCPs that were included.

Further research could be performed to see the effect of the implementation of the clinical pharmacist. It would also be interesting to follow adherence to enteral feeding guidelines [41] or further explore challenges and risks associated with solubility/absorption/formulation.

## 5. Conclusions

Although regulatory actions were taken a long time ago to improve the availability of appropriate medications for children, we still see high numbers of manipulations in this study. It is also alarming to find that health care professionals lack resources such as time and information to support the quality of the manipulations. These findings entail risks of suboptimal absorption/pharmacokinetics, inaccurate doses and is putting the paediatric patient’s safety at risk. We suggest that more ready-to-administer medications be made available through the pharmaceutical industry and compounding companies/pharmacies using traditional or, for example, pharmacoprinting manufacturing techniques. The implementation of clinical pharmacists in the team can likely minimize the necessity of manipulations and further optimise patients’ outcomes.

## Figures and Tables

**Figure 1 pharmaceutics-13-01676-f001:**
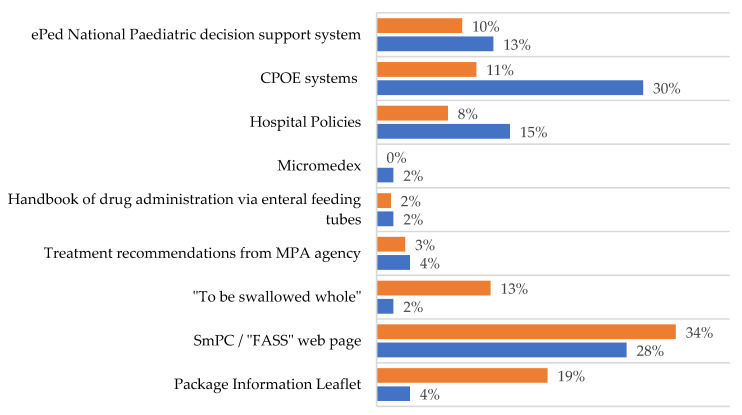
Sources used by paediatricians (blue bars) and nurses (orange bars) to provide additional information in order to ensure that the manipulation procedure is correctly performed. FASS = National web page based on Summary of Product Characteristics [16], *“To be swallowed whole”* is a guide from the pharmacy company “Apoteket AB” [17], March 2013, no longer updated, Treatment recommendations from the Swedish Medical Products Agency (MPA) [18], Handbook of drug administration via enteral feeding tubes [19], CPOE systems include Cosmic, Cytodose and MetaVision, ePed stands for experience and evidence-based database for paediatric medicines. It is a decision support system for drug treatment with various substances and dosage forms for children [20].

**Figure 2 pharmaceutics-13-01676-f002:**
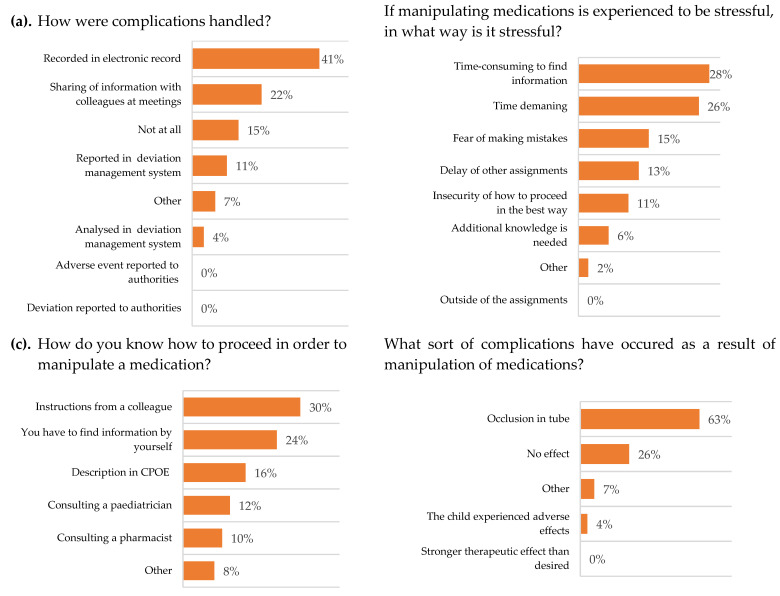
Respondent paediatric oncology and neurology nurses’ experiences and attitudes, N (%). Multiple choices possible. (**a**) *n* = 27, (**b**) *n* = 47, (**c**) *n* = 50 and (**d**) *n* = 27. CPOE = computerized physician order entry software.

**Table 1 pharmaceutics-13-01676-t001:** Patient characteristics and prevalence of manipulations in paediatric oncology ward (A) and paediatric neurology ward (B).

Patient Characteristics	Ward A *n* (%)	Ward B *n* (%)
Included Day ^1^ Patients	232	166
Excluded (<1.5 years)	10 (4%)	15 (9%)
Excluded (No oral meds)	21 (9%)	22 (13%)
Qualified Day ^1^ Patients	201	129
Unique Patients	49	30
Female	88 (43.8%)	55 (42.6%)
Male	113 (56.2%)	74 (57.4%)
Toddlers (1.5–2 years)	3 (1.5%)	12 (9.3%)
Pre-School (2–5 years)	46 (22.9%)	60 (46.5%)
Schoolchildren (6–11 years)	45 (22.4%)	35 (27.1%)
Adolescents (12–18 years)	107 (53.2%)	22 (17.1%)
Age (years), median (min/max)	12.9 (1.6–18.1)	5.8 (1.4–17.2)
Weight (kg), median (min/max)	38.8 (9.2–109)	17 (6.7–105)
Enteral feeding tube present	53 (26.4%)	40 (31%)
Self administration ^2^	24 (11.9%)	13 (10.1%)
Living in Region Uppsala	70 (34.8%)	70 (54.3%)
**Medications Orders**	**Ward A** ** *n* ** **(%)**	**Ward B** ** *n* ** **(%)**
Number of active orders (excluding PRN etcetera)	1271	771
Number of orders for oral delivery (% of active)	791 (62%)	488 (63%)
Administered oral doses	895	463
To Toddlers (1.5–2 years)	2 (0.2%)	23 (5%)
To Pre-School (2–5 years)	11 (1.2%)	98 (21.2%)
To Schoolchildren (6–11 years)	65 (7.3%)	204 (44.1%)
To Adolescents (12–18 years)	817 (91.3%)	138 (29.8%)
Manipulated	117 (13.1%)	94 (20.3%)
-due to inappropriate strength	97 (10.8%)	42 (9.1%)
-due to tube administration	20 (2.2%)	52 (11.2%)
for Toddlers (1.5–2 years)	2 (100%)	14 (60.9%)
for Pre-School (2–5 years)	3 (27.3%)	54 (55.1%)
for Schoolchildren (6–11 years)	25 (38.5%)	11 (5.4%)
for Adolescents (12–18 years)	87 (10.6%)	15 (10.9%)

^1^ Patients were included based on study day. Patients frequently recurred the next study day. For number of unique patients see separate line. Abbreviation PRN means “when necessary/pro re nata”. ^2^ Some patients self-administer only selected oral medications.

**Table 2 pharmaceutics-13-01676-t002:** Manipulations of oral medications in ward A (Oncology) and B (Neurology) including Summary of Product Characteristics (SmPC).

ATC	Active	Dosage Form	Part	Product	SmPC Comment	Ward
A02BC05	Esomeprazole	Sachet	0.5	Nexium 10 mg	Missing info	B
A06AD65	Macrogol combination	Sachet	0.5	Movicol	Missing info	A/B
A12AA06	Calcium	Effervescent	0.5	Calcium-Sandoz	Information regarding divisibility is missing	A
B02AA02	Tranexamic acid	Effervescent	0.5	Cyklokapron 1 g	Information regarding divisibility is missing	A
C07AB02	Metoprolol	MR * tablet	0.5	Metoprolol orion 50 mg	Information regarding divisibility is missing	A
C08CA02	Felodipine	MR * tablet	0.5	Felodipin TEVA	Cannot be divided	B
C09AA02	Enalapril	Tablet	0.5	Enalapril krka 5 mg	Supported. Can be split in two halves	A
H02AB06	Prednisolone	Tablet	0.5	Prednisolon 10 mg	Supported. Can be split in two halves	A
H02AB09	Hydrocortisone	Tablet	0.25 + 0.75	Hydrokortison 10 mg	Can only be split in half	A
H03AA01	Levothyroxine	Tablet	0.5	Levaxin 25 µg	Should not be divided	A
J01DB05	Cefadroxil	Tablet	0.25	Cefadroxil Sandoz 1 g	Information regarding divisibility is missing	A
J05AB01	Aciclovir	Tablet	0.5	Aciclovir 200 mg	Missing info	A
L01BB02	Mercaptopurine	Tablet	0.75	Puri-nethol 50 mg	Cannot be divided. Cannot be crushed.	A
L01BB03	Tioguanine	Tablet	0.5	Lanvis 40 mg	Can be divided only to ease swallowing	A
L04AA10	Sirolimus	Tablet	0.5	Rapamune 1 mg	Cannot be divided. Cannot be crushed.	A
M03BX01	Baclofen	Tablet	0.5	Baklofen alternova 10 mg	Supported. Can be split in two halves	A
M03BX01	Baclofen	Tablet	0.25	Baklofen 25 mg	Can only be split in half	B
M03BX01	Baclofen	Tablet	0.5	Lioresal 10 mg	Information regarding divisibility is missing	B
N02AA01	Morphine	Tablet	0.5	Morfin meda 10 mg	Can be divided only to ease swallowing	A
N02AA01	Morphine	Tablet	0.5	Morfin Alternova 10 mg	Can be divided into two equal parts.	B
N02BE01	Paracetamol	Tablet	0.5	Alvedon 500 mg	Supported. Can be split in two halves	A
N03AB02	Phenytoin	Tablet	0.75 + 0.5	Fenantoin Meda 100 mg	Can be divided only to ease swallowing	B
N03AF02	Oxcarbazepine	Tablet	0.5	Trileptal 600 mg	Can be divided only to ease swallowing	B
N03AG01	Valproic acid	MR * capsule	0.5	Orfiril long 150 mg	Can be opened, but not possible to get accurate dose	B
N03AG04	Vigabatrin	Sachet	0.5	Sabrilex 500 mg	Missing info	B
N03AX11	Topiramate	Tablet	0.5	Topiramat 1A Farma	To be swallowed whole	B
N05BA09	Clobazam	Tablet	0.75 + 0.5	Frisium Sanofi 10 mg	Information regarding divisibility is missing	B
N05CF01	Zopiclone	Tablet	0.5	Imovane 5 mg	Information regarding divisibility is missing	A
N06AA09	Amitriptyline	Tablet	0.5	Amitriptylin 10 mg	Swallowed whole with half a glass of water	A
N07BC02	Methadone	Tablet	0.5	Metadon Abcur 5 mg	Supported. Can be split in two halves	A

* MR = modified release.

**Table 3 pharmaceutics-13-01676-t003:** Summary of findings from ward observations.

Accuracy of Dose	When preparing ¼ of a tablet, the tablet is divided twice using tablet splitter and eye measurement, making the result inaccurate with uneven, crumbly parts. Another approach for preparing ¼ of a tablet was to dissolve the tablet in 10 mL of water and administer 2.5 mL to the patient. When questioned if the nurse reflected upon solubility when dissolving a tablet or a sachet in water and extracting a fraction of it, the nurse did not have a clear answer. There were scales in both ward medication rooms, but they were not used for purpose of, e.g., dividing sachet content due to lack of precision.
Cross Contamination	The splitting/crushing devices were seldom cleaned properly, largely due to the construction of the device. Crushing of tablets containing hazardous medications was performed in safety cabinets and afterwards handled in closed containers (e.g., oral/enteral syringes).
Disposing of unused medication	When only a part of a tablet or sachet was used, the rest was discarded because there were no appropriate stability data or suitable containers.
Ease of Administration	One of the nurses had noticed positive effects of using tablet coating devices (e.g., the Medcoat^®^ product). Some children have a hard time swallowing tablets but may succeed if they are allowed to hold the tablet in the mouth for a while prior to swallowing, which is facilitated by the flavoured coating. Crushing of tablets is done for patients (both with and without enteral feeding tubes) in order to allow for easy swallowing. One nurse had noted an adverse effect when mixing medication with gruel, eventuating a dislike towards the gruel, which is unfortunate, since it is often an essential part of paediatric nutrition.

**Table 4 pharmaceutics-13-01676-t004:** Responding consultant paediatric oncology and neurology paediatricians’ experiences and attitudes.

Experience/Attitude	Number	Never	Rarely	Occasionally	Often	Always	No Opinion
Does the health care provider ensure that the order can be compliant with the regulations ^1^?	*n* = 17 N (%)	0 (0)	2 (11.8)	3 (17.6)	9 (52.9)	1 (5.9)	2 (11.8)
How often do you order medications on paper because of an insufficient CPOE software?	*n* = 16 N (%)	5 (31.3)	9 (56.3)	2 (12.5)	0 (0)	0 (0)	0 (0)
Do you have the time to search for information regarding procedure/instructions for manipulation of a dosage form (off-label)?	*n* = 17 N (%)	3 (17.6)	9 (52.9)	3 (17.6)	2 (11.8)	0 (0)	0 (0)
Do you have the time to write thorough instructions for the procedure of the manipulation of a dosage form (off-label)?	*n* = 17 N (%)	4 (23.5)	5 (29.4)	6 (35.3)	2 (11.8)	0 (0)	0 (0)
**Experience/Attitude**	**Number**	**Strongly Agree**	**Somewhat Agree**	**Neutral**	**Somewhat Disagree**	**Strongly Disagree**	**No Opinion**
The available EHR/CPOE is sufficient for safe medication orders	*n* = 16 N (%)	1 (6.3)	6 (37.5)	6 (37.5)	2 (12.5)	1 (6.3)	0 (0)
**Experience/Attitude**	**Number**	**Difficult**	**Intermediate Difficult**	**Neutral**	**Intermediate Easy**	**Easy**	**No Opinion**
What is your perception of finding information in order to write a detailed medication order in regard to the procedure of manipulation of medication to children?	*n* = 17 N (%)	4 (23.5)	4 (23.5)	5 (29.4)	3 (17.6)	0 (0)	1 (5.9)
**Experience/Attitude**	**Number**	**Yes, Based on:**			**No**	**Other**	**No Opinion**
		**BW**	**BSA**	**BW^0.75^**			
When prescribing medications with unavailable dosing information to children, do you scale down from adult dosages to a corresponding dose suitable for children? If yes, with which scaling model?	*n* = 21 multiple choice N (%)	8 (38.1)	4 (19)	1 (4.8)	2 (9.5)	6 (28.6)	0 (0)

^1^ HSLF chapter 6, §3, see reference to national medication handling regulations from National Board of Health and Welfare [15]. Translated to English: “The caregiver must ensure that the person who prescribes (orders) medication to a child is given the conditions to do so based on the child’s needs: In order to be able to prescribe medications based on the child’s needs, child-specific decision support and such software should be used that make it possible to enter information about; the child’s weight, the child’s body surface, the age of the child, the strength of the diluted medication, and the maximum dose of the medication. The CPOE should also provide conditions for the person prescribing a medication to a child to make a reasonable assessment of the dosage. For continuous and intermittent infusions, the unit of time should be indicated in the dosing instructions in the software, e.g., in the form of mg/body weight/unit of time.” BW = Body weight, BSA = Body Surface Area.

**Table 5 pharmaceutics-13-01676-t005:** Respondent paediatric oncology and neurology nurses’ experiences and attitudes. MR = modified release tablet.

Experience/Attitude	Number	Never	Rarely	Occasionally	Often	Always	No Opinion
Do you feel safe/secure/comfortable with manipulation of medications?	*n* = 21 N (%)	0 (0)	0 (0)	9 (42.9)	12 (57.1)	0 (0)	0 (0)
Do you feel safe regarding working environment/health (for instance if a medication is hazardous)?	*n* = 21 N (%)	0 (0)	2 (9.5)	10 (47.6)	7 (33.3)	2 (9.5)	0 (0)
How often do you have to manipulate the dosage form of a medication for oral administration?	*n* = 21 N (%)	0 (0)	3 (14.3)	8 (38.1)	9 (42.9)	1 (4.8)	0 (0)
How often do you experience patients receiving complications from manipulated dosage forms?	*n* = 21 N (%)	2 (9.5)	12 (57.1)	1 (4.8)	0 (0)	0 (0)	6 (28.6)
If you have experienced complications, were there any serious consequences?	*n* = 13 N (%)	9 (69.2)	4 (30.8)	0 (0)	0 (0)	0 (0)	0 (0)
How often are adverse effects from manipulations reported?	*n* = 18 N (%)	2 (11.1)	10 (55.6)	3 (16.7)	3 (16.7)	0 (0)	0 (0)
How often are clear instructions for execution of the manipulation of medications available in the paediatrician’s order?	*n* = 21 N (%)	5 (23.8)	8 (38.1)	4 (19)	3 (14.3)	0 (0)	1 (4.8)
How often do you consult with the prescribing paediatrician regarding the execution of a manipulation of a medication?	*n* = 21 N (%)	0 (0)	8 (38.1)	9 (42.9)	4 (19)	0 (0)	0 (0)
Do you have the time to search for information on the procedure for the manipulation? (e.g if it is possible to crush/divide a tablet/MR tablet/soft gelatine capsules)?	*n* = 21 N (%)	1 (4.8)	3 (14.3)	7 (33.3)	8 (38.1)	2 (9.5)	0 (0)
How often is it stressful to execute a manipulation of a medication?	*n* = 21 N (%)	0 (0)	4 (19)	9 (42.9)	8 (38.1)	0 (0)	0 (0)
Do you have time to execute manipulations of a dosage form?	*n* = 21 N (%)	0 (0)	1 (4.8)	9 (42.9)	9 (42.9)	2 (9.5)	0 (0)
Would it be valuable to receive ready-to-administer (RTA) medications from the pharmacy?	*n* = 21 N (%)	0 (0)	0 (0)	1 (4.8)	4 (19)	15 (71.4)	1 (4.8)
Do you get questions from the caregiver of the child (parent) regarding manipulating of dosage forms?	*n* = 21 N (%)	9 (42.9)	11 (52.4)	0 (0)	1 (4.8)	0 (0)	0 (0)
Are medications discarded (if only a fraction of a tablet was used) or solutions discarded due to short shelf life after being prepared?	*n* = 21 N (%)	0 (0)	1 (4.8)	1 (4.8)	4 (19)	15 (71.4)	0 (0)

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
