# Peer review of "Manipulations of Oral Medications in Paediatric Neurology and Oncology Care at a Swedish University Hospital: Health Professionals’ Attitudes and Sources of Information"

_pharmaceutics, 2021, doi:10.3390/pharmaceutics13101676_

Round 1

Reviewer 1 Report

The authors have responded adequately to the remarks in the original review.

My only remaining remark regards the conclusion, for which I would propose to change the formulation slightly.  line 397 A clinical pharmacist joint to the team can help searching for ready to use alternatives,  analysing drug interactions, offer advice on proper manipulation and further optimise patients’ outcomes. 

Reviewer 2 Report

The paper represents valuable addition to the current knowledge on necessary manipulation in pediatric medicine administration.

The authors revised the manuscript significantly.

The paper is well written and I suggest it being accepted in the current form.

Reviewer 3 Report

Dear autors

thank you for the revised manuscript. I have no further comments to add. If interested, you might take note of a initiative started in Switzerland regarding dose finding in pediatric patients. https://pededose.ch/de/

Kind regards,

This manuscript is a resubmission of an earlier submission. The following is a list of the peer review reports and author responses from that submission.

Round 1

Reviewer 1 Report

Dear authors

Thank you for this important manuscript. Individual dosing of durgs in pedeatrition divisions are a important issue to overcome. The manuscript is well written and highlights issues in manipulation and distribution of drugs, eg crushing tablets to be provided by a feeding tube.

My comments are meant to enhance the overview of provieded information and improve the interprofessional approach which is necessary to provide efficient and safe care to children.

I wonder why simplifactions are not discussed as an intervention that could be easaly provided by a clinical pharmacist (e.g. the use of solutions instead of tablets containing morphine). Further more, there are clear handling proposals to ensure the patient's and healthcare provider's safety (eg. https://pubmed.ncbi.nlm.nih.gov/31752291)

I'm kind of worried to read that the prescribing pediatritions have not time to ensure the proper administration. In this case it should not be them to provide the final approval for a therapy start but a healthcare professional with the expertise needed (e.g. clinical pharmacist).

I do not agree that the implementation of pharmaceutical care through clinical pharmacists is not able to reduce harm in the clinical setting. There is not enough data to provide such a statement for the pedatritons unit. This is also the key outcome of the provided reference. I further disagree in the tendency the authors discuss their result. They should revise conclusion regarding pharmacoprinting of medication is the one and only solution to overcome the issues they present, since this technique was not part of the study. And there are other issues to overcome regarding the safe and efficient use of this technique.

Reviewer 2 Report

General comments:

Please make sure that abbreviations are explained when used in the main text for the first time.

Specific comments

Please indicate in the abstract the time frame when the study was conducted. In abstract - instead of general sentence : “The working environment of nurses (hazardous substances and stress) and the effectiveness of the medication order/administration software is further discussed” please give more information about actual findings of the study.

Setting:

Please give more info about the hospital where the study was conducted. Why were these two exact wards selected for the study?

Methods:

How was data about the prevalence of manipulation protected to ensure patient anonimity?

Describe Cosmic software in more detail - what data (relevant to the study) can be extracted?

ATC codes are not mentioned anywhere before they appear in the results.

It should be mentioned that this classification will be used in data analysis.

Part about word observation needs more detail. How exactly was that performed? How did the researcher gather data, what type of data was collected?

Results:

Figure 1 would look better as a bar chart ( some of the abbreviations could then be avoided).

Table 4 - to large in the present form and difficult to follow. This should be rearranged in different manner.

The last paragraph of discussion should give more information about pharmacoprinting, because it is one of the main suggestions in the conclusion on how to address issues raised in the paper.

The first paragraph in the conclusion seems out of place in the current form. It needs to be modified, I suggest removing the bullet points and describing the main idea of the paper in a different way.

If possible avoid abbreviations in the conclusion.

Reviewer 3 Report

This is an observational descriptive study on het need for manipulation of medication in order to enable administration to paediatric patients in an oncology and a neurology ward of a tertiary care hospital. As patients' safety is at stake, it is an important subject. It needs however thorough revision before being ready for publication.

The authors use "oral" medication, where also administration through tubes is included. Please use the term "enteral" and "oral" appropriately (throughout the paper).

The introduction focuses on the difficulties of prescribing per kg body weight, without taking into account the available formulations and dosages, and suggest that computerized prescriptions facilitates the ocurence of this problem (line 36-39) without referencing. Please reformulate.

The research question is not entirely clear to me: are manipulations necessary in order to be able to administer medication (because of unability to swallow tablets, or because of administration through a tube), or in order to be able to give the right dose? Were both considered? If so, please differentiate. 

Did you study whether other formulations were available on the market, enabling proper dosing? (e.g. of valproic acid). Were differences in manipulation observed between different nurses administering the same medicine? Or between the same nurse on different days? I do not understand what the authors want to say in line 49-51: activities based on a broad approach?

Line 54 is again difficult to read: is the prevalence of manipulations studied?(or de prevalence of manipulated dosage forms?).

Line 64: were only manipulations aimed at obtaining a fraction of the dose considered? (see comment on research question). Do I understand well that the observational study comprised only one day in each ward? How were manipulations in the first part of the study recorded? selfreporting? Were the nurses prepared for the study? Were the data collected prospectively or retrospectively? The results are very confusingly presented. Please start by mentioning the number of patients in each ward, their age category, and the number of days observed per patient. What is the relevance to mention the total amount of prescriptions, including non-active ones?

Table 1 contains both mean (with SD) and median (and range) for age and for body weight. Median and range is sufficient as the data are clearly not normally distributed. Further on no comparison is made between age categories (or body weight categories) while one would expect more manipulations in the younger age category. Administration of medication through tubes, is a different topic, with own guidelines. There is no information as to whether these guidelines were followed (were medications mixed during administration? how about tube rinsing?).

It is not clear whether Table 2 is extensive, mentioning all medication manipulations and whether are not these are mentioned in the SmPC. Some of the medications mentioned in the list exist in a liquid form (e.g. valproic acid, sirolimus,...) enabling proper dosing, this should be mentioned.

Part 3.2, line 140: how often were tablets crushed?

The list in Table 3 is interesting for a quality improvement project on a ward, but how is it relevant for others (some of the problems mentioned are obvious, such as contamination in a crushing device)?

The results of the questionnaire can be summarized, Table 4 and 5 can be published as supplementary material, but occupy too much space for their message.

The discussion needs thorough revision. It should be focused on the results of this study, relate to what is found in other comparable studies, discuss strenghts and weaknesses, and indicate directions for further research and solutions. The discussion starts with mentioning problems with hand-over, which is not the subject of this study and does not refer to the results. The second paragraph mentions the hazards of self-administration of medication, but again, fails to refer to the own results (did the author observe differences in manipulations between selfadministered of nurse-administered medication?). By offering a liquid formulation whenever available in case of dosages different from tablet doses, manipulation can be avoided.

The role of the hospital pharmacist is not appropriately discussed. There are multiple examples where the clinical pharmacist is systematically involved in ward rounds and surveys the medication in order to avoid interactions, and to minimize the necessity of manipulations. This has a strong educational effect on both nurses and physicians. The clinical pharmacist has the time and competences to answer questions regarding manipulations or administration problems (where physician mention lack of time). The only solution mentioned (3D printing) is not discussed properly with pro's and con's.